# Ozone Disinfection for Elimination of Bacteria and Degradation of SARS-CoV2 RNA for Medical Environments

**DOI:** 10.3390/genes14010085

**Published:** 2022-12-28

**Authors:** Craig Westover, Savlatjon Rahmatulloev, David Danko, Evan E. Afshin, Niamh B. O’Hara, Rachid Ounit, Daniela Bezdan, Christopher E. Mason

**Affiliations:** 1Department of Physiology and Biophysics, Weill Cornell Medical College, New York, NY 10065, USA; 2The HRH Prince Alwaleed Bin Talal Bin Abdulaziz Alsaud Institute for Computational Biomedicine, Weill Cornell Medical College, New York, NY 10065, USA; 3Jacobs Technion-Cornell Institute, Cornell Tech, New York, NY 10044, USA; 4The World Quant Initiative for Quantitative Prediction, Weill Cornell Medicine, New York, NY 10021, USA; 5Department of Computer Science & Engineering, University of California, Riverside, CA 92521, USA; 6Institute of Medical Genetics and Applied Genomics, University of Tübingen, 72074 Tübingen, Germany; 7NGS Competence Center Tübingen (NCCT), University of Tübingen, 72074 Tübingen, Germany; 8Yuri GmbH, 88074 Meckenbeuren, Germany; 9The Feil Family Brain and Mind Research Institute, New York, NY 10065, USA

**Keywords:** microbiome, SARS-CoV-2, hospital-acquired infections, CFU, ozone, disinfection, RNA degradation

## Abstract

Pathogenic bacteria and viruses in medical environments can lead to treatment complications and hospital-acquired infections. Current disinfection protocols do not address hard-to-access areas or may be beyond line-of-sight treatment, such as with ultraviolet radiation. The COVID-19 pandemic further underscores the demand for reliable and effective disinfection methods to sterilize a wide array of surfaces and to keep up with the supply of personal protective equipment (PPE). We tested the efficacy of Sani Sport ozone devices to treat hospital equipment and surfaces for killing *Escherichia coli*, *Enterococcus faecalis*, *Bacillus subtilis*, and *Deinococcus radiodurans* by assessing Colony Forming Units (CFUs) after 30 min, 1 h, and 2 h of ozone treatment. Further gene expression analysis was conducted on live *E. coli K12* immediately post treatment to understand the oxidative damage stress response transcriptome profile. Ozone treatment was also used to degrade synthetic severe acute respiratory syndrome coronavirus 2 (SARS-CoV-2) RNA as assessed by qPCR CT values. We observed significant and rapid killing of medically relevant and environmental bacteria across four surfaces (blankets, catheter, remotes, and syringes) within 30 min, and up to a 99% reduction in viable bacteria at the end of 2 h treatment cycles. RNA-seq analysis of *E. coli K12* revealed 447 differentially expressed genes in response to ozone treatment and an enrichment for oxidative stress response and related pathways. RNA degradation of synthetic SARS-CoV-2 RNA was seen an hour into ozone treatment as compared to non-treated controls, and a non-replicative form of the virus was shown to have significant RNA degradation at 30 min. These results show the strong promise of ozone treatment of surfaces for reducing the risk of hospital-acquired infections and as a method for degradation of SARS-CoV-2 RNA.

## 1. Introduction

The World Health Organization (WHO) has recently stated that we are entering a “post-antibiotic era” after decades of overuse of antibiotics for therapeutic and agricultural reasons [1]. Antimicrobial resistance (AMR) has been increasing at an alarming rate, whereby prevalent nosocomial infections such as pneumonia, tuberculosis, methicillin-resistant *Staphylococcus aureus* (MRSA), and *Clostridoides difficile* infection are becoming difficult to treat with traditional methods, due to multi-drug resistance (MDR) and AMR. In the United States alone, hospital-acquired infections (HAIs) kill an average of 63,000 patients yearly, with many showing AMR. To combat this crisis, resources have been allocated to developing modified versions of existing antibiotics or discovering new ones. However, resistance to new and existing drugs continues to persist due to the strong selective pressure of antibiotics, with successful bacteria acquiring antimicrobial resistance genes in a continual ‘arms race’ between antibiotic development and antibiotic resistance [2,3].

As such, there is a need for novel approaches for effective disinfection tools that control drug-resistant pathogens and reduce antibiotic utilization and consumption. In the past, ozone has been utilized to safely sanitize several products in various industries, such as sewage treatment to kill harmful bacteria [4]. The strong electronegative property of ozone encourages the disruption of proteins, peptidoglycans, and lipids in the cell wall and cell membrane. It interferes with the activity of enzymes and nucleic acids. It has been demonstrated that ozone is up to 3000 times faster acting and 150 times stronger than chlorine for killing bacteria, fungi, and other pathogens under some conditions [5]. Additionally, ozone has been used at highly toxic concentrations to sanitize hospital rooms prior to patient occupation [6]. However, ozone is inherently unstable as it degrades to diatomic oxygen if its production source is exhausted [5]. Thus, a method for continuous ozone generation may be an ideal tool for sanitizing various hospital surfaces or equipment as an alternative to autoclave or ultraviolet light. Here, we report the impact of an ozone-generating machine on medically relevant surfaces and materials from a hospital to gauge the impact on bacterial growth (colony forming units or CFUs) and response (percent of bacteria killed and gene expression analysis).

We examined the bactericidal effect of 20 ppm of continuously generated Ozone safely in an enclosed Sanisport cabinet equipped with an ozone purification system following ozone generation as this concentration can cause tissue damage of the respiratory tract in humans if left exposed [7,8]. Of the bacterial species chosen for this experiment, *E. faecalis* and *E. coli* were studied due to their prevalence in HAI and known relevant burden on hospital systems globally due to their antimicrobial resistance [9,10]. *B. subtilis* and *D. radiodurans* were selected in this study due to their known resistance to DNA damaging agents due to either spore coat assembly as is the case with *B. subtilis* or well protected oxidative damage proteome defense and repair mechanisms as is the case with *D. radiodurans* [11,12]. As *E. coli K12* is a well curated model organism, we further analyzed its stress response to ozone treatment in comparison to no treatment control and hydrogen peroxide treatment via gene expression analysis. Understanding the global oxidative stress regulatory response through transcriptomic changes can help in elucidating protective mechanisms that may lead bacteria to survive and persist in future cases where ozone treatment may lead to resistance phenotypes [13]. This can be useful in preemptively targeting parts of the stress response cascade in ozone and H_2_O_2_ resistant bacteria while also providing the basis for hypothesis driven validation of gene functions and stress response mechanisms [14].

Ozone disinfection of surfaces and hospital personal protective equipment (PPE) has been suggested to be an eco-friendlier disinfectant with a short half-life, leaving no chemical byproducts behind. It has been demonstrated that standard N95 mask filters can withstand many cycles of ozone gas treatment with no degradation of material or filtration efficiency even up to 200 parts per million (ppm) for 90 min or 20 ppm for up to 36 h; both dosages well beyond the proposed viricidal dose of 10 ppm for 30 min necessary for 99% viral inactivation [15,16]. In addition, gaseous ozone can spread throughout a room and has been shown to penetrate multiple crevices and surfaces compared to liquid sprays or aerosols to kill different types of viruses, including aerosol-borne viruses [17].

The severe acute respiratory syndrome coronavirus-2 (SARS-CoV-2) is an enveloped RNA virus that makes it susceptible to ozone degradation of its membrane receptors and protein functional groups as ozone is known to interact with envelope and capsid proteins via the formation of protein hydroxides and protein hydroperoxides [18,19]. Unsaturated lipids, carbohydrates, and nucleic acids are also damaged by ozone, with damage to nucleic acids being one of the primary causes of viral inactivation [19,20]. 

In addition to testing the bactericidal efficacy of ozone we also set out to examine the efficacy of viral RNA degradation from ozone and H_2_O_2_ using Twist Synthetic SARS-CoV-2 RNA Control 2 (MN908947.3) and RNA from a protein-coated and lipid bilayer enclosed non-replicative recombinant form of the virus via the AccuPlex™ SARS-CoV-2 Verification Panel. RT-qPCR demonstrated the degree of ozone and H_2_O_2_ mediated viral RNA degradation at different time points compared to non-treated samples [20,21,22].

## 2. Methods

### 2.1. Bacterial Strains

We used four common HAI-related bacterial species to treat with ozone: *E. coli* strain K12 (substrain MG1655/ATCC 700926), *E. faecalis* (OG1RF ATCC 47077), *B. subtilis* (subtilis strain 168 ATCC 23857), and one isolated strain *D. radiodurans* (R1) provided by the Stewart Shuman lab at Memorial Sloan Kettering Cancer Center. We chose to inoculate surfaces with these species as representative HAI and potentially oxidative stress resistant species as opposed to swab surfaces of hospitals since it is estimated that less than 2% of bacteria can be cultured in a laboratory [23] and for the scope of this study we aimed to examine the bactericidal effect of ozone via culturable CFU counts. 

### 2.2. Ozone Treatment of Bacteria

From freshly grown stock solutions, 100 uL of cultured bacteria were obtained and placed in 900 uL of appropriate broth (tryptic soy broth or nutrient broth with 1% glucose) in 1.5 mL Eppendorf tubes. Bacteria were then serially diluted to 1:100, 1:1000, 1:10,000, 1:100,000, 1:1,000,000, and 1:10,000,000. 100 uL from each dilution was then plated on corresponding agar plates and grown at 37 °C overnight for 12 h. Serial dilutions that produced plates with about 300 CFUs were chosen for the Sani Sport ozone experiment to be able to distinguish between individual colonies.

1 mL of each strain of bacteria was placed in biological triplicates in a 12-well plate for each set of time points 30 min, 60 min, and 120 min. As a positive control, triplicates of each bacteria species were placed in 12 well plates and set aside on a bench top for the duration of the experiment at room temperature. 12-well plates were placed uncovered inside of the Sani Sport cabinet and after each time point of continual 20 ppm ozone exposure, 100 uL of bacteria were collected and plated on appropriate agar plates according to ATCC specific culture conditions for each strain. Plates were incubated at 37 °C overnight for 12 h and CFUs were counted. Plates containing only 100 uL of media were used as negative controls. 

100 uL of each strain of bacteria were also pipetted onto commonly encountered hospital surfaces (syringe, catheter, hospital blanket, and remote controls) in triplicates to be treated with ozone at above mentioned time points. After each time point, the surfaces were swabbed for 3 min and placed in 100 uL of corresponding culture media in 1.5 mL Eppendorf tubes, vortexed for 10 s, and centrifuged at 6000 rpm for 10 min at room temperature to collect bacteria at the bottom of the tube. Bacteria were resuspended in the media and plated on corresponding agar plates at 37 °C overnight for 12 h and CFUs were counted.

### 2.3. E. coli K12 Viability Based on ATP Presence

The effect of ozone on E. coli K12 viability immediately after treatment was assessed using the BacTiter-Glo^™^ Microbial Cell Viability Assay kit (Promega, Madison, WI). Briefly, 10^3^ CFU/mL bacteria were suspended in LB culture medium in a 12 well plate and were subjected to ozone treatment following the above mentioned time points. 100 uL of cell suspension was then transferred to a black-walled 96 well plate in biological quadruplicates. Positive controls were not subjected to ozone and negative controls were prepared using 100 uL of LB culture medium without cells in order to account for background fluoresence. After 5 min of incubation of samples and controls with BacTiter-Glo^™^ reagent, luminescence was measured using a Promega^™^ GlowMax Dicover® plate reader.

### 2.4. RNA Sequencing 

Log growth phase *E. coli K12* was plated in triplicates for each time point on 6 well plates and subjected to ozone for 30 min, 1 h, and 2 h. 100 uM of hydrogen peroxide was used to treat biological triplicates of *E. coli K12* as well for 1 h before RNA extraction. Non-treated controls were set aside for immediate extraction. Immediately after treatment, Roughly 1 × 10^8^ cells were transferred to microcentrifuge tubes and spun down to pellet, and media was aspirated before the cell pellet was treated with Trizol^™^ Max^™^ Bacterial RNA isolation kit. Ribosomal RNA was depleted using NEBNext® rRNA Depletion Kit for bacteria before libraries were prepped using NEBNext® Ultra II kit and adapter-ligated with single Index Primer set 1 NEBNext® Multiplex Oligos for Illumina®. Libraries were then sequenced at 4 million paired-end 150 bp reads on an iSeq100.

### 2.5. Differential Gene Expression and Gene Ontology Analysis

All reads were mapped to the *E. coli* K12 substrain MG1655 reference genome downloaded from Bacteria ENSEMBL using the STAR (ver 2.7) Aligner [24]. The Bioconductor package EdgeR [25] was used to normalize data and perform differential expression analysis between treated samples and non-treated controls with the |Log_2_FC| > 1.5 and adjusted *p* value < 0.05 thresholds. The results were visualized in R with a hierarchical clustered heatmap generated for the top 30 differentially expressed genes, and volcano plots displaying top upregulated and downregulated genes using above thresholds. Gene Ontology analysis for the differentially expressed genes was conducted using TopGo [26]. Fisher analysis was used to test each GO category independently along with a Classic Kolmogorov–Smirnov test for enrichment for 24 terms scored with *p* < 0.01.

### 2.6. Ozone Treatment of Synthetic SARS-CoV-2 RNA

10 uL aliquots of Twist Synthetic SARS-CoV-2 RNA Control 2 (MN908947.3) (1,000,000 copies/uL, 11 pg/uL) were subjected to ozone treatments at 30 min, 1 h, 2 h, 3 h, and 4 h in open capped PCR RNase free PCR tubes in the Sani Sport Supreme. A 0 min control that was immediately processed for first strand cDNA synthesis after thawing on ice from −80 °C was used as the baseline Ct value from which no degradation occurred. Degradation controls were also included in the experiment in which 10 uLs of Twist Synthetic Sars CoV2 RNA was subjected to 1 uL of 100 mg/mL of RNAse A for 30 min at room temperature before being converted to cDNA. No treatment open air controls were also included to compare the degradation rate between normal bench top RNA degradation and degradation from ozone. All treatments and controls were performed in triplicates, and cDNA synthesis was initiated at the end of each time point using Invitrogen SuperScript IV Reverse Transcriptase.

cDNA was then quantified using Qubit RNA HS Assay kit on a Qubit fluorometer to obtain cDNA concentrations of roughly 0.5 ng/uL. 5 uL of cDNA was then added to Luna® Universal Probe One-Step Reaction Mix supplied at a 2X concentration containing Hot-Start Taq DNA Polymerase, dNTPs, and buffer for qPCR on a QuantStudio 6 Flex Real-Time PCR system. 1.5 uL each of a stock 22.5 nmol, CDC 2019-nCoV-N1 Combined Primer/Probe Mix and 2019-nCoV_N2 Combined Primer/Probe Mix^29^ were added to the qPCR reaction with nuclease-free water for a total reaction volume of 20 uL in a 384 clear well PCR plate. Nontemplate controls of nuclease-free water were included to assess the presence of contamination or primer dimers. A standard curve was generated using 1:100 dilutions of stock Twist Synthetic SARS-CoV-2 RNA Control 2 for 10^7^, 10^5^, 10^3^, and 10^1^. All reactions were done in triplicates. A standard Taqman Reaction protocol was set on the Quantstudio 6 with FAM reporter dye selected and no quencher. qPCR conditions consisted of an initial denaturation step of 95 °C for 3 min followed by 45 cycles of a 95 °C denaturation step for 15 s and an extension step of 58 °C for 30 s.

### 2.7. Ozone Treatment of Non-Replicative Virus

To measure the effect of ozone on intact viral particles, we subjected triplicates of non-replicative recombinant viruses via the AccuPlex™ SARS-CoV-2 verification panel to the same experimental conditions as our Twist Synthetic RNA experiment. 100 uL of every 100,000 copies/mL, 10,000 copies/mL, and 1000 copies/mL were aliquoted into PCR tubes and subjected to ozone at 30 min, 1 h, 2 h, 3 h, and 4 h. Controls include a 0 min time point and a 100 uM H202 1 h treatment. All three concentrations were used in order to establish the limits of detection for our modified RNA degradation qPCR assay.

At the end of each time point, treatment RNA from non-replicative virus samples was obtained using the Zymo Quick-RNA Viral kit according to the manufacturer’s instructions. Briefly, 100 uL of 2× Concentrate DNA/RNA Shield™ was added directly to each sample. 400 uL of Viral RNA Buffer was added to the mixture and then transferred to a Zymo-Spin™ IC Column in a collection tube and centrifuged for 2 min at 16,000× *g*. The Column was then transferred to a new collection tube, and 500 uL of RNA Wash buffer was added to the column and centrifuged for 30 s at 16,000× *g* twice, followed by 500 uL of a 95% ethanol wash step in which the column in a new collection tube was centrifuged for 1 min at 16,000× *g*. RNA was eluted in 6 uL of nuclease-free water and then was immediately subjected to cDNA synthesis and qPCR as mentioned above.

### 2.8. Statistical Analysis

Each bacterial and viral RNA experiment was carried out with three biological replicates. The CFUs obtained from the bacteria replicate experiments were subjected to paired Student’s *t*-test after normality was assessed using the R function Shapiro–Wilk test. *p*-values were obtained by comparing no treatment groups at 0 h for each corresponding bacterial species, time point, and surface. Significance codes 0.001 ***, 0.01 **, 0.05 *.

For viral RNA, ozone treated replicate Ct values were subtracted from the 0 h no treatment control to obtain delta Ct values and then log transformed using 2^−∆Ct^. The means of the log transformed delta Ct values were then subjected to one-way analysis of variance and Tukey’s method was used to distinguish *p*-values for multiple comparisons. Global *p* < 2.2 × 10^−16^. Significance codes 0 ***, 0.001 **, 0.01 *.

## 3. Results

### 3.1. Bacteria Susceptibility to Ozone Treatment

We tested four distinct bacterial species commonly found in the hospital environment (*E. coli* strain K12, *E. faecalis*, *B. subtilis*, and *D. radiodurans*) for their susceptibility to ozone. These bacteria were grown and tested in triplicate on four high-traffic surfaces from the hospital equipment: catheters, blankets, hospital remote controls, and syringes, with positive and negative controls, also included for comparison. All samples were treated with the same ozone dose at increasing lengths of time (30, 60, and 120 min), and CFU counts were compared between ozone-treated and control groups. Control experiments consisted of allowing bacteria to grow without ozone treatment and then collecting samples as outlined in the Methods for each of the time points. Each surface and species tested colony forming units (CFUs) were counted, CFU/mL calculated, and the triplicates were averaged with the standard error of the mean calculated to obtain ozone kill curves (Figure 1). These results showed as much as three-log-fold changes in CFUs at the 1 h and 2 h time points, with most cases of of CFU counts halved within the the first 30 min of treatment.

Moreover, both the different surfaces and different species showed distinct rates of reduction and response to the treatment. For example, *E. coli* showed the greatest sensitivity to killing on the 12-well plate but less so on the remote controls. The surface with the most species killed was the syringe, specifically at the 2 h time point (>99.3%). While hospital blankets showed the greatest variance (Figure 2). The 2 h time points for 12-well plate experiments showed a wide range of bacterial reduction across the different species, from 89.75% to 99.70%. The syringe experiments were the most consistent, with a range of bacterial decrease from 94.24% to 99.57%; conversely, the other surfaces showed a more comprehensive range of bacterial reduction, from 83.59% to 96.00% for the catheter, 86.10% to 99.31% for the remote control, and 82.99% to 98.36% for the hospital blankets. These data demonstrate that the ozone treatment causes significant reductions in all bacterial species tested across the various surfaces (*p* < 0.05, Student’s *t*-test), with increasing degrees of efficacy as a function of time.

### 3.2. Gene Expression Profiling of Ozone Treatment

To ensure gene transcripts from live bacteria were sequenced, cell viability of *E. coli K12* was assessed immediately after ozone treatment via presence of ATP luminescent signal as CFU counts do not provide insight into bacterial death right away. No significant difference in viability was seen for the 30 min and 1 h time point but at 2 h there was a significant increase in ATP presence (*p* < 0.01 Student’s paired *t*-test) (Figure 3a). Hierarchical clustering of gene transcripts revealed tight clustering of stress response genes at each time point in *E. coli K12* in response to ozone treatment, with 263 genes downregulated and 184 upregulated as determined by a |Log2FC| > 1.5 (Figure 3b). Many of these top-upregulated expressed genes were typical of *E. coli K12′s* response to oxidative damage, such as the reactive oxygen species (ROS) scavengers *SodA*, *AhpC*, and *KatG* [27]. Other genes of interest include the DNA binding protein *Dps*, which acts as a physical protectant of DNA during stress response but also collects iron to act as a reducing agent under oxidative stress [28]. Heat shock response genes such as *DnaK*, *IbpB*, and *IbpA* were also shown to be upregulated [29]. Differential gene expression for 2 h of ozone treatment vs. no treatment were also visualized with a volcano plot using|Log2FC|> 1.5 and p cutoff = 0.05. Here, we observed an upregulation of heat shock response genes *DnaK* and *IbpA* [30], and a downregulation of DNA oxidative damage response gene *Dps* [31] (Figure 3c). A volcano plot for the comparison of 1 h of ozone treatment to 1 h of H_2_O_2_ treatment revealed key differences in stress response gene activation between the two types of oxidative damage such as the upregulation of ferrous iron transport genes *feoA* and *feoB* [30], cytochrome oxidase *cydA* [32], and again heat shock response and protein repair gene *DnaK* [29] (Figure 3d). 

We further analyzed some of the top differentially expressed genes induced by ozone treatment at all time points by organizing them into functional Gene Ontological categories. Ranking of annotated genes into GO biological processes based on Fisher’s test (*p* < 0.01) and Classic Kolmogorov–Smirnov (*p* < 0.01) revealed an enrichment of terms for oxidative-reduction processes, cellular respiration, and tricarboxylic acid metabolism to name a few (Table 1).

### 3.3. SARS-CoV-2 RNA Susceptibility to Ozone Treatment

Synthetic SARS-CoV-2 RNA was shown to have significant degradation 1 h into ozone treatment (*p* < 0.001) using one-way ANOVA Analysis and Tukey’s method (global *p* < 2.2 × 10^−16^) among comparisons of cycle threshold (Ct) values obtained from SuperScript IV Reverse Transcriptase qPCR (Table 2). While some non-significant degradation of RNA occurred in non-treated controls left to air exposure on a benchtop, degradation was significant for samples after 1 h of treatment, with percent RNA log fold change of 4 h time points nearing complete RNA degradation as compared to RNAse treated samples. Control samples treated with air exhibited an RNA Fold change of 98.69%, 93.55%, 86.22%, 61.78%, and 50.42% of intact amplifiable RNA remaining for 30 min, 1 h, 2 h, 3 h, and 4 h, respectively, with 0 h control representing 100% of RNA; In contrast, ozone treated samples exhibited 65.13%, 25.82%, 11.24%, 12.46% and 6.16% of RNA remaining for 30 min, 1 h, 2 h, 3 h, and 4 h. RNAse treated samples represent complete degradation with 0% intact amplifiable RNA remaining (Figure 4a).

RNA from non-replicative capsid enclosed SARS-CoV-2 showed significant degradation 30 min into ozone treatment as could be reliably detected by 10,000 copies of virus (*p* < 0.001) and 1000 copies of virus (*p* < 0.001) (Table 2). 100 uM of H_2_O_2_ treatment for 1 h also produced significant degradation according to Ct values obtained compared to 0 h treatment (*p* < 0.001 for both 10,000 and 1000 copies of virus), and RNA degradation was still comparable to 1 h of ozone treatment, *p* = 0.99. For 10,000 copies of the virus, the percent of log fold intact amplifiable treated RNA left over compared to 0 h control treatment representing 100% of RNA was 19.33%, 15.72%, 4.03%, 2.34%, 0.49% for 30 min, 1 h, 2 h, 3 h, and 4 h of ozone treatment, respectively. 1 h of H_2_O_2_-treated RNA left over was 18.02%, and RNase-treated samples represented thoroughly degraded RNA with 0% RNA left (Figure 4b). One thousand copies of the virus showed similar results to the 10,000 copies of the virus tested. The percent RNA log fold change of ozone-treated samples compared to 0 h was 25.58%, 5.46%, 3.43%, 1.61%, and 1.01% for 30 min, 1 h, 2 h, 3 h, and 4 h of ozone treatment, respectively. H_2_O_2_ treated samples were 28.06% degraded after, and RNase representing complete degradation was 0% (Figure 4c). Mean Ct values were obtained along with Ct standard deviation. To obtain changes in degradation, delta Ct values were calculated as either treatment or timepoint mean Ct subtracted from 0min control mean Ct. From this, RNA fold-change vs. control was calculated as 2^−∆Ct^ as the difference between one cycle reflects a 2-fold difference in starting transcript level [33,34].

## 4. Discussion

Antibiotic resistance is a major financial burden on the United States healthcare system, often associated with common nosocomial infections which are caused by the sharing of rooms of infected patients, transmission by hospital workers, overgrowth of pathogens in the patient’s own microbiome, and interactions with surfaces and equipment that harbor these pathogenic bacteria. Fortunately, a variety of disinfection methods are being developed to tackle this looming challenge.

A recent study by Rangel and colleagues sought out to test the efficacy of ozone as a disinfection method among a set of antibiotic resistant bacterial species as a means of addressing HAI. Here, they exposed bacteria in a room measuring 38 m^3^ to ozone at low doses ranging from 0.6 to 2.1 ppm for 10 and 12 h but found that these low concentrations led to limited or inconclusive data [35]. Our findings support that ozone treatment is an effective sterilization method to combat HAIs in medical environments, but this may be due to using a higher dose in a contained environment, rather than diffuse ozone at acceptable OSHA and EPA levels throughout a room, thus highlighting the utility of a Sani sport cabinet system. We report the rapid killing of the majority of medically relevant bacteria within 30 min, and up to a 99% reduction in viable bacteria at the end of 2 h treatment cycles, with as much as a 3–4-fold log kill range. Here, we used the Sani Sport Supreme Dupliskate Ozone generator, but without any modifications or changes to the instrument. As such, changes in the flow rate, pressure, humidity, or temperature could increase the efficacy of this method and possibly reduce bacterial species on hospital surfaces in less time [36,37].

It should be stressed here that the use of a closed cabinet system for ozone generation of 20 ppm for this duration is necessary as the current OSHA permissible exposure limit is 0.1 ppm over eight hours [7,38]. Ozone is a powerful oxidant that upon exposure can lead to bronchial inflammation and hyper-responsiveness that leads to airway obstruction and diminishing lung function over time. There is a growing body of evidence that short term exposure can lead to chronic obstructive pulmonary disease (COPD) hospitalization as well as increased incidences of asthma [8]. In an EPA systematic review of ozone effects on lung function from 2013 to 2020, it was demonstrated that even at very low concentrations of ozone, children and the elderly suffered from diminishing lung function [39].

We demonstrated that different surfaces led to variation in the reduction of the same bacterial species. For example, the syringe and catheter experiments killed off bacteria at a slower rate than remote control and 12-well plate experiments. A likely explanation for this is that bacteria were inoculated directly onto the surfaces of 12-well plates and remote controls so that ozone could interact with more bacteria without any obstruction. Regardless, ozone treatment still demonstrated bacterial reductions even when inoculated inside tube-like structures such as catheters and syringes. Furthermore, ozone may not damage sensitive equipment (vs. bleach), ensuring the continued use of sterilized equipment in medical environments. Further work could explore the mechanisms by which bacteria respond to this treatment. Ozone is thought to disrupt membrane integrity, so monitoring viability in this context can provide a more conservative methodology [5,35]. While utilizing CFUs as a method to assess bacterial reduction has been the gold standard in many kill curve experiments, this method is limited in being able to discriminate viable, but not cultivable cells (VBNC), and other methods such as propidium monoazide (PMA)-qPCR or flow cytometry could also be used [40]. In the context of further testing more pathogenic bacteria known to enter the VBNC state, it would be worthwhile to use these methods in conjunction with colony-forming unit counts. As we observed little to no changes in viability immediately after ozone exposure via presence of ATP, it would be necessary to test the immediacy of ozone’s bactericidal action, as one of the major limitations of this study is that it took 12 h to finally obtain CFU counts. Without this information it would be difficult to ascertain just how soon a piece of equipment is considered sterile. 

It is interesting to note that ozone kills *D. radiodurans*, as this species has been known to have the unique ability to reconstruct its fragmented genome in response to ionizing radiation [41]. There are several theories of how *D. radiodurans* is able to survive such extreme stressors, one of them being an unusual capacity to avoid radiation-induced protein oxidation. Thus, it would be interesting to study the stress response pathways in the context of Sani Sport ozone-mediated treatment [42,43,44]. Understanding these mechanisms can be valuable for discovering novel genes for a multitude of bacterial species to decipher phenotypic characteristics, virulence regulation, and survivability, which may impact other medical environments like ambulances [45] and broader urban environments around the world [46].

While our methods demonstrate that 20 ppm of ozone can severely reduce several bacterial populations, we wanted to further explore gene expression changes in this context. The utilization of these stress responses is typically mediated by global regulatory mechanisms, which affect biochemical pathways leading to physiological changes that confer survival. As expected, we found that regulatory networks such as ROS scavengers, cellular respiration, tricarboxylic metabolic processes, and protective DNA damage responses were activated in *E. coli K12* [28,29,31]. Interestingly, Heat shock responses were downregulated as ozone treatment continued up to 2 h, but 1 h of hydrogen peroxide treatment did not induce a downregulation of these genes compared to 1 h of ozone treatment. One possible explanation for this phenomenon is that it has been documented that low levels of hydrogen peroxide treatment can induce a cross-adaptive response to heat shock and ethanol treatment [31]. To our knowledge, no examples of ozone at this dose eliciting cross-adaptive responses exist. All other ROS responses of *E. coli K12* remained comparable between the 1 h of ozone treatment and 1 h of hydrogen peroxide treatment.

Finally, we describe a new method for determining RNA degradation for SARS-CoV-2 utilizing a replication-deficient non-infectious virus that closely resembles the wild-type virus. While RNA degradation has been shown to be one of the main mechanisms for viral inactivation [37,47] to validate these results further, traditional viral plaque assays using infectious SARS-CoV-2 virus must be employed, as synthetic RNA or inactivated virus is not a substitute for live virus. This study would be essential to further validate the efficacy of ozone treatment as we demonstrated that a 99% decrease of 10^6^ copies/mL of RNA would still result in about 10^4^ copies/mL which could still result in infections in plaque assays [48]. In a review by Bayarri and colleagues, genome integrity as well as protein capsid damage and molecular organization were seen as other mechanisms in ozone mediated inactivation in several viruses [36]. Given that SARS-CoV-2 is an enveloped virus it is likely that proteinic and lipidic parts of the envelope could be rendered inactive by ozone according to molecular dynamic simulations [47]. In a recent study utilizing viral plaque assays it was demonstrated that gaseous ozone exposure at 81% relative humidity and provided at a rate of 15 g/m^3^ for 1 h in a 3 L plastic box reactor could inactivate SARS-CoV-2 up to 99% on both glass and stainless steel surfaces [37], although the mechanism from which viral inactivation took place remains to be elucidated. Surprisingly the amount of time it took to reduce RNA copy number to RNase levels was high, a phenomenon indicating stable RNA that was also noted by Kitagawa and colleagues when they subjected live virus to 222 nm UV light [49]. Nonetheless, our method provides an effective way to measure the efficacy of ozone mediated RNA degradation in both naked RNA and capsid enclosed RNA. As previously demonstrated, N95 mask filtration efficiency is resistant to damage via many cycles of treatment or high dosages of ozone [15,16]. It should be noted that there have been several conflicting reports on the persistence of SARS-CoV-2 on varying surfaces. However, A growing body of evidence does suggest that fomite transmission of SARS-CoV2 remains plausible [50,51,52,53]. Thus, ozone can be a viable option for the sterilization of PPE and many different types of surfaces. In a recent study by Torres-Mata and colleagues, ozone treatment at a range of concentrations on various office and clinical supplies proved effective with 90 ppm of ozone treatment at 120 min being optimal for disinfection of large volume supplies. Here, RT-qPCR was also used to assess elimination of SARS-CoV-2 RNA from these surfaces [50]. While our limit of detection using our primers and qPCR chemistry was 1000 copies of the non-infectious virus; this methodology could also be adapted to other primer sets and qPCR kits to perhaps measure even lower amounts of copies of virus per sample [22,54,55].

## 5. Conclusions

In this study, we utilized an ozone generating cabinet to test its efficacy of killing commonly encountered bacteria on various surfaces encountered in hospital settings. Using colony-forming unit counts, we demonstrated significant reductions in bacterial populations across a range of surfaces with increasing levels of disinfection over time. Gene expression profiling showed expected enriched stress response-related genetic signatures in *E. coli K12* in response to ozone treatment that was distinct from hydrogen peroxide treatment, the basis of which can help provide for future hypothesis driven studies resulting in the elucidation of some of the underlying functional mechanisms of oxidative damage resistance in bacteria. We also adapted our ozone treatment methodology to demonstrate the efficacy in degrading both naked synthetic SARS-CoV-2 RNA and enveloped RNA in the form of non-infectious capsids as damage to nucleic acids is one of the mechanisms for which viral inactivation takes place. While not a substitute for actual infectious virus, working with inactivated and synthetic samples can help broaden access to work with SARS-CoV-2 in a manner that mitigates safety concerns while retaining some of the inactivated characteristics of the virus for future use. These results can help guide future investigations into reducing the transmission of HAI add provide a means for disinfection of hospital environments. 

## Figures and Tables

**Figure 1 genes-14-00085-f001:**
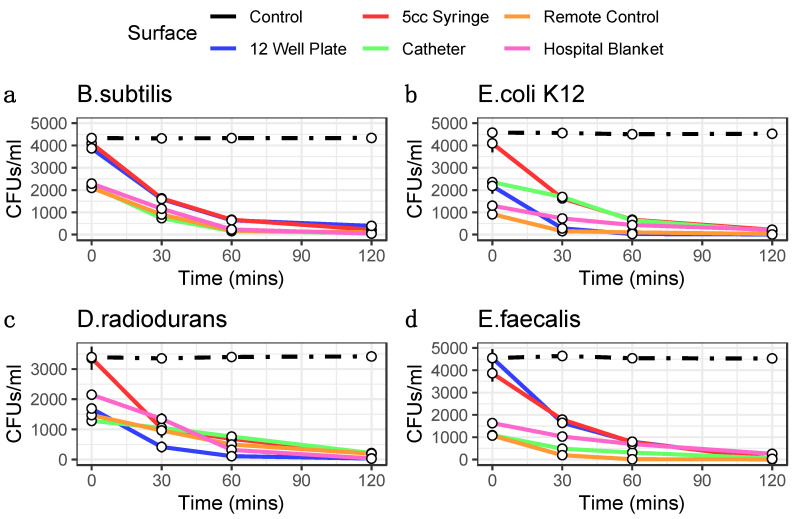
Ozone Kill Curves indicate ozone treatment reduces bacterial load. 1 mL of bacteria for taxa *B. subtilis* (**a**), *E. coli K12* (**b**), *D. radiodurans* (**c**), and *E. faecalis* (**d**) were pipetted into wells of 12 well plates and 100 uL was collected at each time point for plating. 100 uL of bacteria were pipetted inside 5 cc syringes, inside catheters, onto remotes, and onto hospital blankets in triplicates and swabbed for 3 min at each time point for plating. Colony forming units were then counted following plating and 12 h incubation. CFUs are converted into CFUs/mL (*y*-axis). Error bars represent standard error of the mean. Controls are indicated as black lines. Ozone treatment (*x*-axis) is at 20 ppm.

**Figure 2 genes-14-00085-f002:**
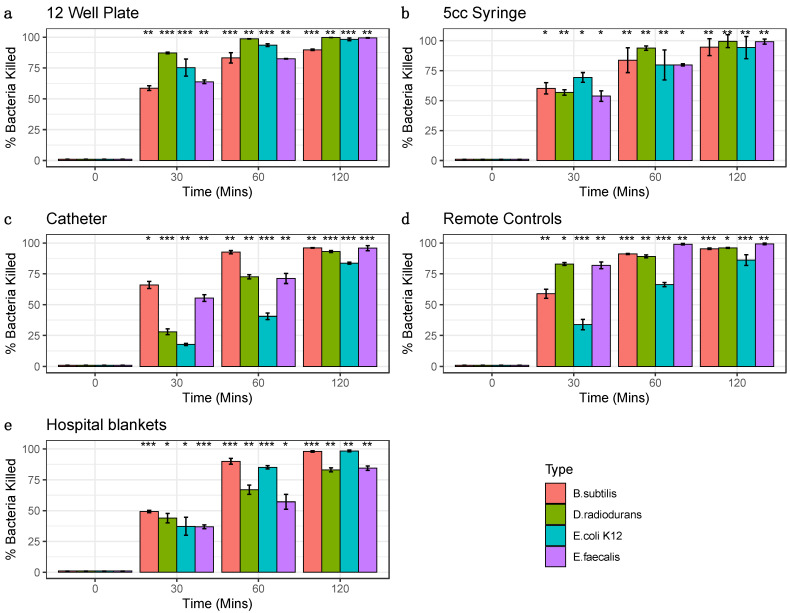
Percent Bacterial Reduction. Bacterial percent reductions for *B. subtilis* (red), *E. coli K12* (green), *D. radiodurans* (blue), and *E. faecalis* (purple) from ozone treatment (20 ppm). Error bars represent standard error of the mean from triplicate experiments on a variety of surfaces: (**a**) 12 well plates (**b**) syringes (**c**) catheters, (**d**) remote controls (**e**) hospital blankets. Bacteria species at each time point were compared to corresponding control 0 min (Significance codes 0.001 ***, 0.01 **, 0.05 *, Student’s paired *t*-test).

**Figure 3 genes-14-00085-f003:**
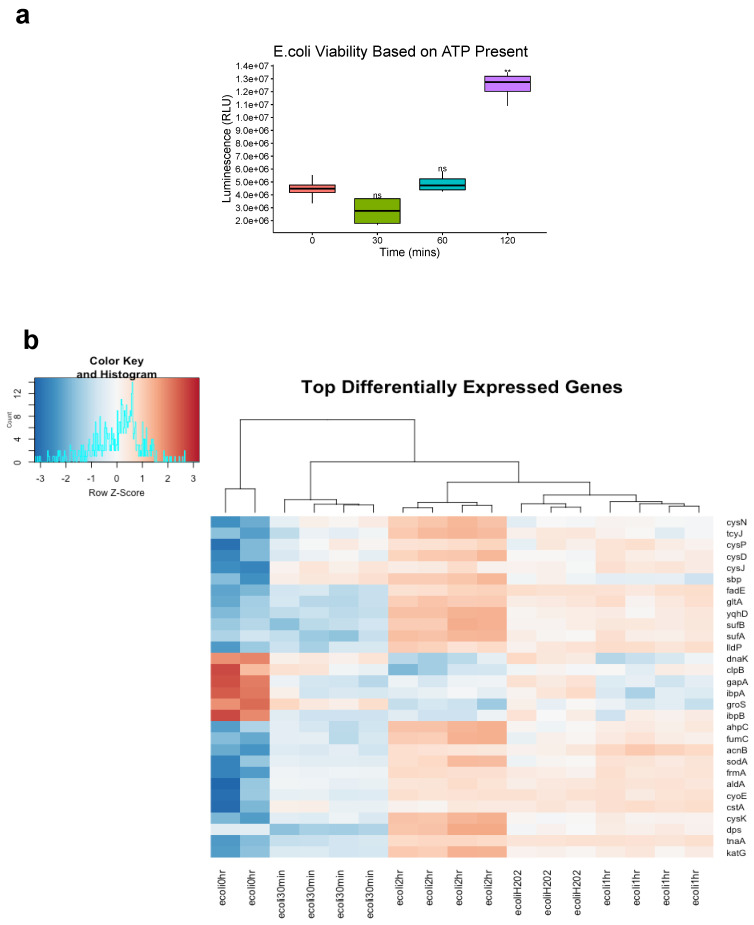
Top differentially expressed *E. coli* K12 Genes. (**a**) E. coli K12 cell viability was assessed by measuring presence of ATP immediately after ozone treatment. ATP signal was measured via luminescence and expressed as Relative Light Units (RLU). Bacteria species at each time point were compared to corresponding control 0 min (Significance codes 0.01 **, Student’s paired *t*-test). (**b**) E. coli K12 at 0 h no treatment were compared to E-coli K12 subjected to 20 ppm of ozone at 30 min, 1 h, and 2 h, along with 100 uM H202 treatment for 1 h. Differential expression fold changes are evaluated based on |Log2FC| > 1.5. 263 genes were downregulated while 184 were upregulated, LogCPM values were selected for the top 30 differentially expressed genes. (**c**) Differential expression was analyzed between 2 h of ozone treatment compared to no treatment control and (**d**) 1 h of ozone treatment compared with 1 h of hydrogen peroxide treatment, LogFC > Log_2_(1.5) and p cutoff = 0.05.

**Figure 4 genes-14-00085-f004:**
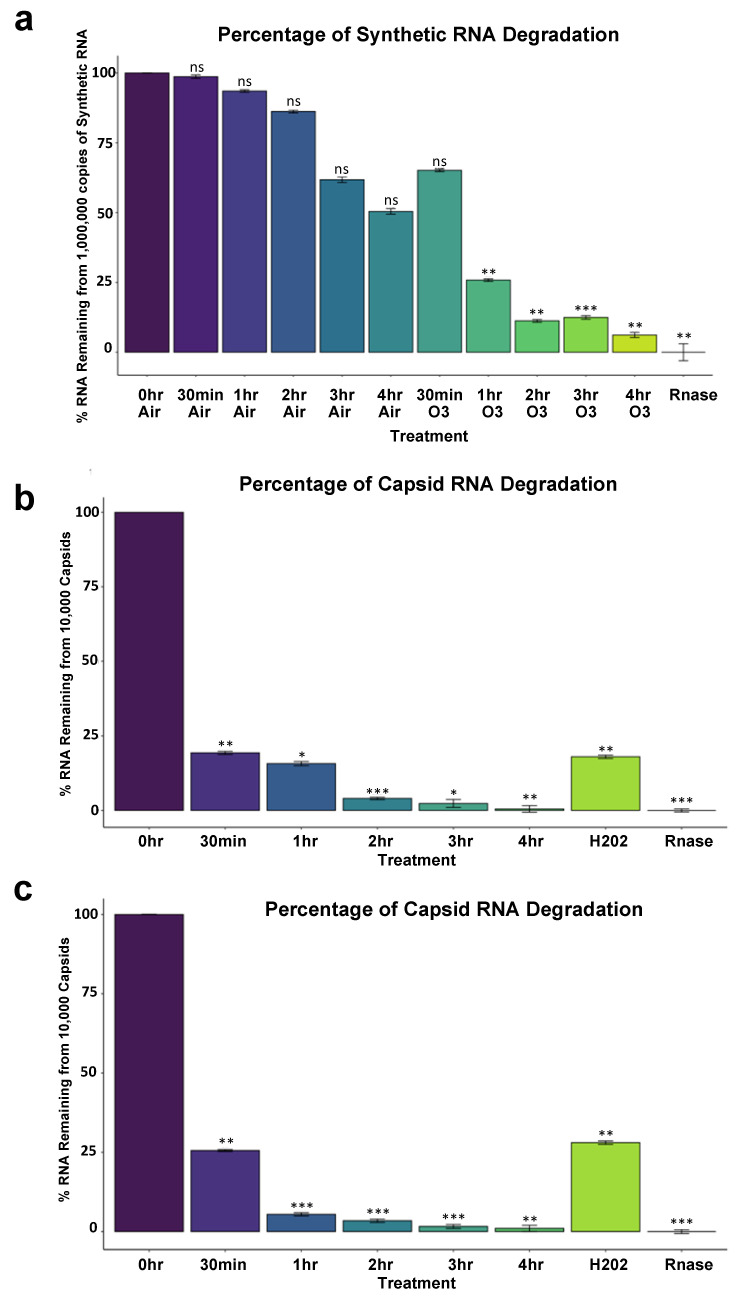
SARS-CoV-2 percent log fold change between ozone treatments and controls. (**a**) 10,000,000 copies of Synthetic SARS-CoV-2. (**b**) Ten thousand copies of non-replicative SARS-CoV-2. Global *p* < 3.2 × 10^−13^. (**c**) One thousand copies of non-replicative SARS-CoV-2. 0 h no treatment control was used as the comparison reference for ozone-treated and control samples left on the benchtop exposed to air. Rnase values indicate maximum RNA degradation. Treatment Ct values were first subtracted from 0 h control to obtain delta Ct values (ΔCT = CT target − CT reference). Delta Ct was then transformed into a log scale using 2^−∆Ct^. Error bars represent variance and was calculated as s = (s12 + s22)^1/2, s = standard deviation. One-way ANOVA Analysis computed global *p*-values, and Tukey’s method was used to distinguish *p*-values for multiple comparisons. Global *p* < 2.2 × 10^−16^. Significance codes 0 ***, 0.001 **, 0.01 *.

**Table 1 genes-14-00085-t001:** Top 10 Significant GO Terms. Enrichment analysis on top GO terms for biological processes was conducted using Fisher analysis, where each GO category was tested independently. 488 GO terms were scored with 22 terms with *p* < 0.01. Classic Kolmogorov–Smirnov test was also used to test for enrichment, with 24 terms scored with *p* < 0.01. Annotated genes were 948, significant genes were 297, the minimum number of annotated genes to a GO term was 10, and nontrivial nodes were 488.

GO.ID	Term	Rank in Class	Classic Fisher	Classic KS
GO:0055114	oxidation-reduction process	1	8.7 × 10^9^	5.9 × 10^11^
GO:0045333	cellular respiration	2	5.7 × 10^7^	8.4 × 10^9^
GO:0009060	aerobic respiration	3	1.8 × 10^6^	2.2 × 10^8^
GO:0015980	energy derivation by oxidation of organisms	4	2.8 × 10^6^	1.1 × 10^7^
GO:0006099	tricarboxylic acid cycle	5	8.2 × 10^6^	1.7 × 10^6^
GO:0006101	citrate metabolic process	6	8.2 × 10^6^	1.7 × 10^6^
GO:0072350	tricarboxylic acid metabolic process	7	8.2 × 10^6^	1.7 × 10^6^
GO:0006091	generation of precursor metabolites	8	2.9 × 10^5^	2.5 × 10^6^
GO:0072329	monocarboxylic acid catabolic process	9	5.6 × 10^5^	3.0 × 10^5^
GO:0016999	antibiotic metabolic process	10	6.5 × 10^5^	1.3 × 10^5^

**Table 2 genes-14-00085-t002:** RNA Data. Ct values and % RNA fold change across different time points.

**Synthetic RNA**	**Ct Mean**	**Ct SD**	**△△** **CT Treatment-Control**	**% RNA Fold Change vs. Control**
0 h Control	18.22	0.48	0.00	100.0
30 min air exposure	18.24	0.37	0.02	98.7
1 h air exposure	18.32	0.07	0.10	93.5
2 h air exposure	18.43	0.29	0.21	86.2
3 h air exposure	19.99	0.42	1.77	29.3
4 h air exposure	20.56	0.12	2.34	19.7
30 min O3	18.68	0.16	0.46	72.5
1 h O3	18.79	0.14	0.57	67.2
2 h O3	21.37	0.13	3.15	11.2
3 h O3	21.22	0.44	3.00	12.5
4 h O3	22.24	0.78	4.02	6.2
Rnase	38.81	3.01	20.59	0.0
**10,000 copies of** **Capsid**	**Ct Mean**	**Ct SD**	**△△** **CT Treatment-Control**	**% RNA fold change vs. control**
0 h Control	25.26	0.55	0.00	100.00
30 min O3	27.13	0.87	1.86	27.53
1 h O3	27.81	2.74	2.55	17.10
2 h O3	29.31	0.75	4.04	6.07
3 h O3	29.08	2.49	3.82	7.09
4 h O3	28.54	0.67	3.28	10.33
H_2_O_2_	27.57	0.53	2.30	20.29
Rnase	41.40	0.73	16.13	0.00
**1000 copies of** **Capsid**	**Ct Mean**	**Ct SD**	**△△** **CT Treatment-Control**	**% RNA fold change vs. control**
0 h Control	28.46	0.40	0.00	100.00
30 min O3	30.05	0.05	1.59	33.17
1 h O3	32.53	0.37	4.07	5.95
2 h O3	32.73	0.36	4.27	5.19
3 h O3	32.77	1.30	4.31	5.04
4 h O3	33.95	0.51	5.50	2.22
H_2_O_2_	30.27	0.45	1.81	28.45
Rnase	41.40	0.73	12.94	0.01

## Data Availability

All raw data and CFU counts are included in the manuscript.

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
