# Peer review of "Ozone Disinfection for Elimination of Bacteria and Degradation of SARS-CoV2 RNA for Medical Environments"

_genes, 2022, doi:10.3390/genes14010085_

Round 1
Reviewer 1 Report
In this manuscript, Westover et al propose the use of ozone to disinfect surfaces. This Reviewer is concerned about the potential use of this method for several reasons: 1) how can the authors propose the use of something that is harmful to humans when inhaled, without mentioning any precautionary measurements? 2) how can they claim in their title that this treatment ‘eliminates’ SARS-CoV-2 when no experiments with the real virus have been made and 3) how can the authors propose the use of this method on the basis of ‘antibiotic resistance is an issue’? In addition, sequencing on bacteria treated is done – how do the authors can claim the findings when it is likely that they are sequencing dead bacteria, as per their data?
Abstract:
The conclusions section suggest to treat people, not surfaces, as it is written. I hope this is a mistake and please rephrase it as an alternative to disinfect surfaces.
Introduction:
Line 108: more susceptible than non-enveloped viruses? The interactions with proteins will happen regardless of the envelope… If this is a hypothesis by the authors please rephrase as such.
Lines 111 to 117: where is the reference for such a statement, particularly ‘most’? The scale of detection of SARS-CoV-2 varies greatly, and depends on the assay employed. It has also varied greatly during the pandemic depending on the variant tested. To this Reviewer’s knowledge this is incorrect. Quantifyng RNA amounts in swabs is in many cases not possible and having RNA in them does not equate to having virus since there is a lot of host RNA. Please rephrase or delete.
Lines 121-122: I agree, but rather than research, if the virus has to be inactivated laboratories can contribute to testing perhaps. Any research with the virus itself will require Cat-3 facilities. Moreover, how would disinfecting surfaces allow for workers to handle live virus?
Lines 130-131: please provide a reference or data to show that RT with random hexamers + Superscript IV is ‘more robust against partial degradation’.
In the case of viral RNA degradation – how would gel electrophoresis would tell us anything? It can be used as a surrogate for RNA degradation in the sample but the RNAs visualised are 18S and 28S from the host?
In their Abstract the authors explain the results for their experiments in bacteria but there is nothing about that in the Introduction.
Results
The data show that after the first 30 minutes of treatment the CFUs are halved in most cases, not log-reduced.
It is not clear why the authors use a Fisher’s test when they are measuring a difference (t-test or U-test) or even a ratio test. In addition, how did they check for normality of their data?
If bacteria are dying how can the authors test for differential gene expression, particularly at 2h?
The heading ‘SARS-CoV-2 susceptibility to ozone treatment’ is misleading. This should clearly state that the experiments were done with synthetic RNA.
Line 186-187: ‘degradation robust modified qPCR method’ please show data to show that it is degradation robust and differs significantly vs standard one-step qPCR.
The RNA appears to decrease by 50% left in the air 4h, which the authors mention. However they started from 106 copies/mL and even if decreased to 1% of that (decreased by 99%) that means that there are still ~104 copies/mL which can be, in the case of virions, infectious in plaque assays.
I am not sure what the data of the capsid enclosed RNA adds.
Discussion and Conclusions
Antibiotics are mentioned – these are not used for disinfecting surfaces. And on the use of SARS-CoV-2 with potentially degraded RNA – I am not sure that this is of use?
Reviewer 2 Report
At first, I would like to appreciate the opportunity for reviewing this interesting paper. This manuscript needs grammar check and english writing improvement.
Although the paper idea is promissing the manuscript lacks in methods description and the study might need improvement in statistical analysis. I included a point-by-point paper review to Editorand Author's concerning the paper impairments and hoping to contribute with comments to improvement.
Minor Comment
1-It is a great honor to review this manuscript, because I use ozone on a daily base routine.
2-While I was reviewing the manuscript, I discovered a preprint in BioRxiv (https://www.biorxiv.org/content/10.1101/420737v3) would suggest to verify if the journal allows it.
3-I suggest to authors to read the paper https://pubmed.ncbi.nlm.nih.gov/34677149/.
Title
1-I would suggest to modify to “Ozone disinfection strategies forbacteria and SARS-CoV2 elimination among different surfaces”.
Abstract/ keywords
1-The currently abstract is not according as journal authors guideline I would suggest to modify according as journal authors guideline.
2-Where is the methods section in the currently abstract?
3-Page 1, line 34: I would suggest to change “cleaning” for “disinfecting”.
4- The result section is confusing please rewrite it.
5- Why authors cited antibiotics in the conclusion, in my humble opinion it does not meet the study objective and criteria to disinfect areas or environments.
5- Please check the keywords.
Introduction
1-I would suggest to authors rewrite the entire first 2 paragraphs, to me, readers should know about the post antibiotic era, although, authors insist in to correlate ozone disinfection to ozone medications therapy in the introduction, which gives to readers different ideas regarding paper objective.
2-Please add to the end of introduction the paper objective.
3-I suggest to explain better about ozone, how it works for disinfections and its use risks instead to explain about RNA because it is better suited to methods section.
Methods
1-The method section is poorly written and needs major improvement. Some concerns must be written as:
- Which were the inclusion and exclusion criteria? How the study was conducted?
- How bacterias species were selected??
- Why did the authors inoculate bacterias species in hospital surfaces? Was not easier to use this species as control group and collect samples from hospital surfaces prior the cleaning process?
- Why use different types of bacterias and SARS-CoV-2?
- What antibiotic was used to compare with ozone?
- The RNA sequencing is poorly written and has no sense to the methods presented, please improve the text.
Results
1- Please describe better your results, because authors only describe few information then display the tables.
Discussion
1- It is poorly discussed using the authors results against literature results.
Conclusion
1- It is derivative and I recommend to authors to rewrite this section trying to make improvements, mainly because the authors could not affirm that ozone made a difference in disinfect hospital surfaces without comparative parameters.
Round 2
Reviewer 2 Report
authors achieved the all my considerations
Author Response
Thank you, we have updated language style and minor spell check accordingly.